# New Perspectives in Studying Type 1 Diabetes Susceptibility Biomarkers

**DOI:** 10.3390/ijms26073249

**Published:** 2025-03-31

**Authors:** Yongsoo Park, Kyung Soo Ko, Byoung Doo Rhee

**Affiliations:** Department of Internal Medicine, Sanggye Paik Hospital, Inje University College of Medicine, Seoul 01757, Republic of Korea; kskomd@paik.ac.kr (K.S.K.); bdrhee@paik.ac.kr (B.D.R.)

**Keywords:** type 1 diabetes, immune regulation, insulin secretion, β-cell stress

## Abstract

Type 1 diabetes (T1D) is generally viewed as an etiologic subtype of diabetes caused by the autoimmune destruction of the insulin-secreting β-cells. It has been known that autoreactive T cells unfortunately destroy healthy β-cells. However, there has been a notion of etiologic heterogeneity around the world implicating a varying incidence of a non-autoimmune subgroup of T1D related to insulin deficiency associated with decreased β cell mass, in which the β-cell is the key contributor to the disease. Beta cell dysfunction, reduced mass, and apoptosis may lead to insufficient insulin secretion and ultimately to the development of T1D. Interestingly, Korean as well as other ethnic genetic results have also suggested that genes related with insulin deficiency, let alone those of immune regulation, were associated with the risk of T1D in the young. Genes related with insulin secretion may influence the phenotype of diabetes differentially and different genes may be working on different steps of T1D development. Although we admit the consensus that islet autoimmunity is an essential component in the pathogenesis of T1D, however, dysfunction might occur not only in the immune system but also in the β-cells, the defect of which may induce further dysfunction of the immune system. These arguments stem from the fact that the β-cell might be the trigger of an autoimmune response. This emergent view has many parallels with the fact that by their nature and function, β-cells are prone to biosynthetic stress with limited measures for self-defense. Beta cell stress may induce an immune attack that has considerable negative effects on the production of a vital hormone, insulin. If then, both β-cell stress and islet autoimmunity can be harnessed as targets for intervention strategies. This also may explain why immunotherapy at best delays the progression of T1D and suggests the use of alternative therapies to expand β-cells, in combination with immune intervention strategies, to reverse the disease. Future research should extend to further investigate β-cell biology, in addition to studies of immunologic areas, to find appropriate biomarkers of T1D susceptibility. This will help to decipher β-cell characteristics and the factors regulating their function to develop novel therapeutic approaches.

## 1. Introduction

Diabetes mellitus is a heterogeneous group of disorders with hyperglycemia as a common feature. Patients with long-standing hyperglycemia face a high risk for chronic microvascular (nephropathy, retinopathy) and macrovascular (atherosclerosis) complications irrespective of strenuous efforts for glycemic control. Hyperglycemia results from metabolic defects causing deficits in insulin secretion, insulin action, or both. Previously diabetes was classified along therapeutic lines, as either insulin-dependent or non-insulin dependent diabetes. However, the American Diabetes Association (ADA) has recommended an etiological classification (Table 1) [1]. Diabetes resulting from a deficiency of insulin secretion, but not from the deficient insulin action, is classified as type 1 diabetes (T1D). However, it has been generally accepted that T1D refers to immune-mediated diabetes and results from the destruction of the insulin-secreting β-cells of the islets of Langerhans in the pancreas. A distinguishing feature of this autoimmune type 1 diabetes (T1aD) is the presence of anti-islet autoantibodies. However, there are recent thoughts disputing the immunologic pathogenesis of T1D but emphasizing the permissible role of β-cells. T1D was once thought to be a childhood disease but it is now recognized that T1D can present at any age [2].

## 2. Markers of Humoral Autoimmunity

The T1aD is usually associated with chronic and progressive autoimmune destruction of islet β-cells with a long prodromal phase. The autoimmune phenomena associated with T1aD include lymphocytic infiltration of pancreatic islets and circulating serum antibodies to various islet cell antigens. The first test for anti-islet autoantibodies consisted of the determination of antibodies reacting with frozen sections of human pancreas, termed cytoplasmic islet cell autoantibodies (ICAs) [3]. This test has formed the basis for many studies of the natural history of the disease and the basis for several current large clinical trials of diabetes prevention. This test detects a subset of glutamic acid decarboxylase (GAD) 65, insulinoma-associated antigen (IA-2), zinc transporter, and yet to be defined autoantibodies, and fails to detect insulin autoantibodies [4]. Despite the utility of the ICA test, it has been difficult to standardize and has limited predictive potential in the absence of readily measurable ‘biochemically’ detected anti-islet autoantibodies [4,5]. Table 2 lists six islet autoantigens for which there currently exist autoantibody assays based upon recombinant DNA production of the autoantigen [4,5,6]. The β-cell molecules, chromogranin A, islet-specific glucose 6 phosphatase catalytic subunit-related protein (IGRP), and proinsulin are autoantigens in the non-obese diabetic (NOD) mouse and help us to understand the pathogenesis of human T1D. Studies analyzing multiple autoantibodies have been especially important in demonstrating that T1aD is typically characterized by the expression of antibodies to not one, but several islet cell proteins. The results of recent studies indicate that unaffected relatives who express multiple autoantibodies in contrast to those expressing only one are very likely to progress to diabetes [5,6]. Notwithstanding, the significance of specific autoantibodies in the pathogenesis of T1D remains unclear.

## 3. Type 1 Diabetes Is an Immune-Mediated Disorder

Although the diagnosis of T1aD rests on the detection of humoral antibodies directed against β-cell antigens, it is now believed that β-cell autoimmunity is primarily a T-cell-mediated process [7]. This conclusion is based on evidence of the cellular infiltration (insulitis) in islets of NOD mice long before the development of overt diabetes, T-cell adoptive transfer experiments, and isolation of islet-specific T-cell clones [8]. Although it was once thought that GAD was the primary antigen recognized, attention is shifting towards insulin as a potential primary initiating molecule [9]. Insulin/proinsulin is the only β-cell-specific autoantigen identified to date. The conformational insulin epitope which reacts with insulin autoantibodies is defined [10]. The majority of CD4+ T-cell clones isolated from NOD mouse islets are activated by insulin and a specific epitope (amino acids B:9-23) [11]. A proinsulin NOD transgenic mouse is protected from diabetes and insulitis [12]. T cells also target chromogranin A and IGRP, which extend the list of T-cell antigens in T1D. T1D is not at all or rarely developed in the chromogranin A gene knock out NOD mouse [13]. Antigen-specific immunotherapy targeting IGRP could induce antigen-specific CD8+ T-cell-targeted immune regulation and delay development of T1D [14].

The relative roles of CD4+ and CD8+ T-cells in pathogenesis are currently under intense investigation. Studies in transgenic mice indicate that CD8+ T-cells alone are capable of β-cell destruction, but CD4+ islet-specific T-cell clones from the NOD mouse are also capable of inducing diabetes after adoptive transfer [15,16]. Treatment of NOD mice with anti-CD8 antibodies can prevent diabetes if given early in life (2–5 weeks old) but not if given later [17]. The presence of CD8+ clones in islets of young NOD mice also supports the theory that CD8+ cytotoxic cells are important for the initiation of the disease process by cell lysis, with CD4+ cell expansion. Since β-cells do not express HLA class II antigens, specialized antigen presenting cells (dendritic cells and macrophages) must be involved in the activation of the islet specific CD4+ T-cells.

## 4. Type 1 Diabetes Is a Genetic Disease

Unlike several diabetes-related syndromes with a known genetic etiology, T1D is of unknown genetic etiology in the vast majority of patients. Familial aggregation is, however, still apparent among these cases and family studies have been useful for estimating empiric risks to relatives, testing hypotheses relating to genetic subgroups, determining the relative contribution of genetic and environmental factors to individual differences in disease susceptibility, and as first step in identifying disease genes by positional cloning [18,19,20]. The risk for T1D in the first-degree relatives of patients is about 15-fold higher than the disease prevalence of about 0.004 in Caucasian population. About 6% of siblings of patients are affected giving the ratio of the risk for siblings of T1D patients and the population prevalence (λs), often used to assess the degree of familial clustering of a disease is 15 (λs = 0.06/0.004 = 15) [19]. However, the pattern of inheritance is complex, making it difficult to relate recurrence risks in relatives to the segregation of individual genes. Such observations are typically taken as an indication that multiple genes are involved in disease pathogenesis.

Because of the inherent difficulties in achieving unbiased case ascertainment and in collecting and documenting family material, family studies of T1D have not kept pace with the progressive refinements in the phenotype. Most studies have been restricted to twins or first degree relatives. ‘Point’ estimates of risk or concordance in monozygotic (MZ) twins range from 10 to 55% [20]. This wide range of concordance differences may result from differences in the length of follow-up and/or lack of life table analysis. Current life table results have estimated that the lifetime risk is greater than 70% for a MZ co-twin of a patient, and less than 10% for dizygotic (DZ) co-twins or siblings of patients [21]. Islet-cell autoimmunity through the comparison of both progression to diabetes and autoantibody expression in non-diabetic DZ twins and MZ twins of patients with T1D have been evaluated and while MZ twins had a very high risk of progressing to diabetes and anti-islet autoantibodies expression, DZ twins had low progression to diabetes and a low prevalence of anti-islet autoantibody expression [22]. These results suggest that islet cell autoimmunity is also mainly genetically determined.

## 5. Importance of HLA in Determining Genetic Susceptibility

From these twin and other family studies as well as other population studies, we now know that T1D is a genetic disease with complex genetic trait. Around 50 years ago, the HLA, the expression of which might be associated with insulitis [23], was found to contain a major locus that influences predisposition to T1D [24]. The HLA region, also known as the human major histocompatibility complex (MHC), is a cluster of over 150 genes contained in about 3.5 Mb of DNA on the short arm of chromosome 6 (6p21.3). The products of many of these genes play a central role in the immune response and in susceptibility to T1D. Out of these genes, the class II region (termed *IDDM1*) determines the major genetic susceptibility of T1D [25,26,27], which suggests that the selective presentation of specific autoantigen peptides is involved in the pathogenesis [28]. Although there have been some studies that show weaker or no association, several studies have demonstrated that both DQ and DR influence T1D susceptibility [29,30]. It has now become evident that there are both susceptible and protective alleles at DRB1, DQA1, and DQB1 loci. Although the effects of these alleles are the same across different countries, ethnic differences in the prevalence of specific alleles in patients varies with population allele frequencies, which makes a different level of association. Specific DQ and DR alleles are non-randomly associated with each other on what are termed extended haplotypes, the typing of which provide the best risk determinants of T1D. Nearly all studies are consistent in demonstrating positive associations between T1D and DRB1*0401-DQA1*0301-DQB1*0302 and DRB1*0301-DQA1*0501-DQB1*0201, and a negative association between T1D and DRB1*1501-DQA1*0102-DQB1*0602, with the strongest diabetes association seen in the heterozygote for the high-risk alleles [30]. In addition to these two high-risk HLA haplotypes, there are less common HLA haplotypes associated with high diabetes risk, which can be found in Asian ethnicities (Table 3) [31,32,33]. At least some of the 15-fold increased risk of diabetes among patients’ siblings can be attributed to the fact that relatives often share high-risk HLA haplotypes. Based on a collection of studies, about 5–10% of affected siblings inherit different HLA haplotypes from both parents (0 identity by descent). This is considerably lower than the expected frequency of 1/4, thereby confirming the evidence for HLA-linked susceptibility reported in case-control studies. But the contribution of HLA sharing to the increased risk for disease in siblings (measured as the ratio of expected to observed frequencies of siblings with no HLA haplotypes in common) is about 2.5–5.0 (denoted as λ(HLA),s), considerably less than the overall 15-fold increased risk (λs = 15) [34].

In addition to the HLA genes, other polymorphisms within the region might affect T1D disease risk [35]. These include genes associated with antigenic processing within the class II region, such as *LMP*, *TAP*, *HLA-DM*, and *HLA-DO*, as well as genes within the class III region. This class III region is located between class I and class II loci and contains many immunologically important genes, such as those encoding tumor necrosis factor and some complement components in innate immunity. Some evidence suggests that polymorphisms within these regions affect the disease risk conferred by risk-associated DR-DQ haplotypes even after stratifying for HLA-DP and class I markers [36].

## 6. Multiple Genes Outside HLA

In addition to the HLA genes, candidate gene approaches have found the strongest association with T1D within polymorphic genes of the insulin (*INS*) and *PTPN22* (which encodes a lymphocyte protein, tyrosine phosphatase), both of which are associated with reductions in immune tolerance and increased T-cell activation [37,38]. Thereafter, several genome-wide association studies have revealed a number of other loci associated with T1D. Other genes that might influence the T1D risk include *IL2RA* (encoding α subunit of the IL-2 receptor), interleukin genes (mainly *IL-4* and *IL-13*), *PTPN2* (protein tyrosine phosphatase, non-receptor type 2), *IFIH1* (interferon-induced helicase), *BACH2* (basic leucine zipper transcription factor 2), *GLIS3* (Gli-similar 3 protein), and ubiquitin-associated and SH3 domain-containing protein A (*UBASH3A*) genes (Table 4) [39,40]. These new genes tend to have a smaller influence on the risk of T1D [41]. Moreover, in many cases, the gene responsible for the effect has still remained unclear due to the strong linkage disequilibrium (LD) through large genomic areas. In some genes, such as *IL2RA* and *UBASH3A*, multiple single-nucleotide polymorphisms (SNPs) were found to be independently associated with the risk of T1D, and their combinations might show higher risk than single SNPs [42,43]. Interestingly, some newly identified genes are associated with β-cell function, suggesting the important role of β-cell function as well as immune regulation in the pathogenesis of T1D [37,44]. However, genetic susceptibility to T1D can vary significantly across different ethnicities and the importance and strength of individual non-HLA genes may differ. By the way, many of these immune genes are also associated with other autoimmune diseases, including rheumatoid arthritis and autoimmune thyroid disease, denoting the shared mechanisms of different autoimmune diseases [44].

Polymorphisms in the insulin gene associated with T1D can now be typed using SNPs, which have also been associated with different lengths of tandem repeats (variable number tandem repeats (VNTRs)) in the noncoding region [37]. The ectopic expression of various tissue-specific proteins in the thymus is important for the generation of central tolerance, and the disease-associated genotype of the insulin gene is associated with the poor expression of insulin in the thymus, which leads to autoreactive insulin-specific T-cells not being destroyed during the thymic education of T cells [45,46]. Thus, a genetically determined variation in the expression of self T1D antigens in the thymus may affect the shaping of the T-cell repertoire and ultimately, susceptibility to T1D. Moreover, alternative splicing also determines differential IA-2 expression in thymus compared with islets. Islets express full-length IA-2 mRNA and two alternatively spliced transcripts, whereas the thymus exclusively expresses an alternatively spliced transcript lacking exon 13. This exon 13 transcript encodes for the important T1D target epitopes, supporting the concept that tolerance to IA-2 epitopes not expressed in the thymus may not be achieved. This way, differential splicing as a regulatory mechanism of gene expression plays a permissive role in the development of autoimmune responses to IA-2 [47,48]. The importance of the thymus in the development of T-cell tolerance is also well known in autoimmune polyendocrine syndrome type 1 (APS-1), which can also feature T1D. In APS-1, mutations in the autoimmune regulator (*AIRE*) gene result in thymic epithelial cells being unable to present ectopic autologous antigens and induce T-cell tolerance. Inherited mutations in this gene can thus cause a multi-faceted syndrome where T1D is a part of the clinical picture [49].

T1D is also a part of another rare immunological syndrome that is characterized by loss of function of a single gene, *FOXP3*. The FOXP3 deficiency leads to multi-organ autoimmunity syndrome, with immunodysregulation polyendocrinopathy X-linked syndrome having autoimmunity characteristics as well as severe immunodeficiency [50]. The patients with this syndrome have inefficient regulatory T-cell function and the resistance of effector T-cells to the inhibition by regulatory T-cells. No disease-associated polymorphisms in the *FOXP3* have been found in patients with T1D who do not have this syndrome. These rare syndromes thus reveal various defects in immune regulation that are critical for protection from T1D-associated autoimmunity.

Some of the genetic polymorphisms outside HLA that are associated with T1D are nonsynonymous substitutions. As for *PTPN22*, defects in the encoded regulator molecule (LYP) affect the function of immune cell populations (B and T cells) [38]. Another important polymorphic gene associated with T1D is *IFIH1*, which encodes the MDA5 molecule that functions as a receptor of dsRNA generated during replication of enteroviruses and induces type 1 interferon production [51]. Rare nonfunctional mutations in *IFIH1* protect against T1D, which reveals the importance of the inflammatory reaction associated with interferon production for β-cell destruction in pancreatic islets. TYK2 protein is a key mediator in type 1 interferon induction in β-cells and the reduced function of TYK2 protein protects against T1D [52].

Most non-HLA polymorphisms seem to be outside the coding regions and control the expression of genes in their vicinity and those located some distance away or even in different chromosomes. The effects of these polymorphisms are often limited to certain cell subpopulations [53]. Identifying these effects is still demanding and in the early phase, but it has the potential to produce new information on disease mechanisms. By the way, some of these new genetic loci appear to predispose people to T1D independently of HLA and may be important factors in families with T1D who lack strong HLA susceptibility. Other loci may interact to cause susceptibility, and specific combinations may be diabetogenic. The many risk polymorphisms in the loci of both HLA and non-HLA regions most often have quantitative effects, with no single locus resulting in a severe functional defect. A selection of the strongest and best characterized non-HLA loci are listed in Table 4 together with brief details on the suggested mechanisms of action [54]. Thus, the critical balance instability that causes T1D might result from different individual combinations of genetic polymorphisms inducing progressive islet autoimmunity and/or increasing β-cell vulnerability, which can now be defined using a ‘genetic risk score’ [55].

## 7. Type 1 Diabetes in Asia Is Also a Genetic Disease

T1D is much less frequent in Asia than in countries with a predominantly Caucasian population [32]. The ratio of the risk for siblings of Japanese T1D patients with age-at-onset under 20 years and the population prevalence (λs) in Japan was found to be more than 200, a much higher value than that in Caucasian populations [56]. Familial clustering of a disease does not necessarily indicate a genetic component, since it may be caused by the sharing of the same environment among family members. If a genetic factor is responsible for the high λs value for T1D with low population prevalence in Asians, then susceptibility alleles whose frequencies are very low in the general population may be segregating in T1D families [57]. Similar to what is observed in the Caucasian populations, the HLA class II alleles on chromosome 6p21 are also found to be the most highly related to diabetes susceptibility [32,33]. In Asians, both the DR and DQ alleles are the ones to have the highest association with T1D. Table 3 shows the distribution of HLA DR alleles in Korean patients with T1D and control subjects. In the T1D cases, HLA DR3 and DR9 were increased. DR4 as a group was not significantly increased in diabetic patients compared to controls. Among the DR4 subtypes, DRB1*0401 and DRB1*0405 had increased frequencies in patients. Two DR4 subtypes (0403 and 0406) had lower frequencies in patients. DR15 confers strong protection. DR12 was also strongly protective. When the HLA DQB1 alleles were identified in the T1D patients, the only DQB1*0201 allele had significantly higher frequencies in patients, while three DQB1 alleles (0301, 0601, 0602) had significantly lower frequencies in patients compared to controls. Five haplotypes (DRB1*03-DQB1*0201, DRB1*0401-DQB1*0302, DRB1*0405-DQB1*0302, DRB1*0407-DQB1*0302, and DRB1*0901-DQB1*0303) had significantly increased frequencies in diabetic patients. Four other haplotypes (DRB1*15-DQB1*0601, DRB1*15-DQB1*0602, DRB1*08-DQB1*0601, and DRB1*10-DQB1*05) had significantly lower frequencies in patients [54,57].

In addition to HLA genes, Korean genetic results demonstrated that genes related with insulin deficiency as well as those of immune regulators were associated with the risk of T1D [58]. When we stratified T1D patients according to the age of onset, genes related with immune regulation showed significant association with the susceptibility of T1D in the early-onset subgroup, while those of insulin secretion were associated with the late-onset subgroup. Genes related with insulin secretion may influence the phenotype of diabetes differentially and different genes may be working on different steps of T1D development. In fact, the non-HLA susceptibility of other Asian T1D patients might also be the result of genes that induce progressive islet autoimmunity as well as increased β-cell vulnerability [59].

## 8. Adult-Onset Type 1 Diabetes

Although T1D was once thought to be a childhood disease, it is now well recognized that T1D can present at any age. Most recent new cases of T1D appears to occur in adulthood, although statistics on this are not enough to reach a conclusive answer. With increasing age at onset, the clinical course of the disease appears to be milder and β-cell destruction tends to be slower than in childhood-onset T1D, which makes the distinction between T1D and type 2 diabetes (T2D) more challenging [2]. Low serum c-peptide concentration, a marker of severe endogenous insulin deficiency, might be useful to guide both classification and treatment [60]. However, no single clinical feature can perfectly distinguish T1D from other subtypes at diagnosis. Classification depends on an appreciation of other risk factors for T1D and the integration of clinical features (age at diagnosis and body-mass index) with biomarkers (islet autoantibodies) (Figure 1) [61,62]. Similar challenges apply to latent autoimmune diabetes in adults (LADA), which shows both features of T1D and T2D [63,64]. In one study of Korea, although the LADA prevalence estimated by anti-GAD positivity appeared to increase, the true insulin dependency evidenced by multiple antibody positivity did not increase [63]. A unique approach to estimate the frequency of T1D among adulthood diabetics was applied using a genetic risk score for T1D in participants of the UK Biobank [65]. The results suggested that the incidence of new onset T1D is stable even in adulthood in the UK, and the study concluded that 42% of T1D cases were diagnosed in people aged between 31 and 60 years.

## 9. Ambiguous Autoimmune Pathogenesis

Although T1D is believed to be caused by an immune-mediated process at large, there has been some ambiguities of the role of immune cells in the pathogenesis of T1D. In fact, despite stringent negative selection by central tolerance, islet auto-reactive T-cells are very common in the healthy population, and, moreover, most individuals with islet autoantibodies will never develop T1D [66,67]. Most patients with T1D have immune function and regulation that is indistinguishable from those of healthy individuals. Almost all patients with cancer who are treated with immune checkpoint inhibitors do not develop T1D [68,69]. Furthermore, some patients with T1D present with negligible T-cell autoimmunity [70]. Moreover, induction of autoimmune diabetes in mice by treatment with islet T1D autoantigens is very difficult [71]. Transduction of human islet auto-reactive T-cell receptors (TCRs) in humanized mice leads to high frequencies of T-cell autoimmunity to islets, but no diabetes was induced [72]. In addition, progression of T1D has not been found to accelerate after patients with T1D are injected with islet autoantigens [73,74,75]. In contrast, proinsulin expression from gestation until weaning was sufficient to completely protect NOD mice from diabetes, insulitis, and development of insulin autoantibodies. Inducing tolerance to islet antigens in the perinatal period is sufficient to impart lasting protection from diabetes [76]. Autoantigen treatment to prevent T1D in recent onset T1D patients was insufficient in human subjects but preserved residual β-cell function [75]. In addition to the possible minor efficacy of peroral insulin in high-risk individuals, autoantigen prevention trials have failed. Adequate autoantigen selection, dose, and route would be important [74,77]. The aim of β-cell antigen vaccination is to induce tolerance by inducing autoantigen-specific T-regulatory cells, which also downregulate the activity of proinflammatory antigen-presenting cells [73]. The anti-CD3 monoclonal antibody Teplizumab, which was approved by the US Food and Drug Administration to delay the progression of T1D in patients at high risk of developing T1D. Several mechanisms of targeting CD3 with anti-CD3 monoclonal antibodies have been postulated including direct inactivation of T cells, inhibition of apoptosis induction and induction of apoptosis or anergy of activated T-cells. While these new therapies show promise, achieving long-lasting control of blood sugar levels and the possibility of stopping insulin use remains difficult [77]. Furthermore, immunotherapies targeting cytokines in T1D have not yet proven a considerable effect on disease protection [78,79]. These ambiguities and inconsistencies in our understanding of the critical role of islet autoimmunity need to be further explanation.

## 10. T1D as a β-Cell Disease

Normal β cell physiology includes a high demand for insulin production and secretion in response to dynamic glucose sensing. This secretory function predisposes β cells to significantly higher levels of endoplasmic reticulum (ER) stress compared to non-secretory cells. In addition, many environmental triggers associated with T1D onset further augment this inherent ER stress in β cells. Therefore, people with T1D have reduced β-cell function at diagnosis compared with healthy controls. Before diagnosis, decreased first-phase insulin response (FPIR) during intravenous glucose tolerance testing (IVGTT) is an early indicator of β-cell dysfunction and predictor of T1D [80]. In the TrialNet and DPT-1 studies, c-peptide index as well as some oral glucose tolerance test (OGTT) measures depicting reduced β-cell function were found to serve as an FPIR alternative in their ability to predict T1D in autoantibody positive subjects [81,82]. Over time, the function of many of these residual cells gradually deteriorates. However, analysis of the pancreatic sections from patients with long-term T1D reveals the presence of residual β-cells [83,84]. Utilizing c-peptide concentration as a parameter of β-cell function, 30–80% of people with long-term T1D are found to be insulin microsecretors [85,86]. Thus, although endogenous β-cell quantity and function decline with long duration of T1D, some patients do not progress to a complete loss of all β-cells. The presence of residual β-cell function after the diagnosis of T1D increases the possibility of an improved effect of interventions targeted at rescuing or augmenting the survival of this residual pool of β-cells. However, analyses of pancreatic specimens from the Network of Pancreatic Organ Donors repository have not found any evidence of either increased neogenesis or proliferation in pancreatic cells from donors with T1D [84]. Therefore, true mechanisms underlying the persistence of residual β-cells in some patients with long-term T1D remain unclear. Identifying pathways that have allowed these cells to escape the autoimmune attack could give us insight into new therapeutic approaches.

By the way, considering the fact that auto-reactive T cells are part of a normal T-cell repertoire and immune regulation in T1D patients as a whole is not different from controls, it is unlikely that the T1D is entirely a result of defects of dysfunctional immune cells. Instead, it could be the result of a defective local environment favoring the peripheral activation of the immune system in the targeted tissue. As early as 1960, Bottazzo proposed a crucial role of β-cells in determining their own destiny in the natural course of T1D [87]. β-cell abnormalities might contribute to autoimmune pathogenesis leading to the notion of ‘β-cell suicide’. Various different triggers including viral infection and/or metabolic stress might change the size and function of the pancreatic β-cells and lead β-cells to elicit an immune response [88]. Although β-cells do not express HLA class II antigens, HLA class I over-expression is common in pancreatic sections from cadaveric donors with T1D. This over-expression serves as a homing signal for cytotoxic CD8+ T lymphocytes [89]. However, this signal might be a primary β-cell defect or a response to various stimuli such as viral infection. Additionally, evidence also exists for increased β-cell ER stress associated with accelerated β-cell death [90,91]. ER stress in β-cells has been associated with alterations in mRNA splicing and errors in protein translation and folding, which results in peculiar protein products of potential immunogenic neoantigens [92].

In fact, islet transplantation, a treatment for T1D, has met significant challenges since a substantial fraction of the islet mass fails to engraft, partly due to death by inflammation in the peri- and post-transplantation periods. As ER stress is important in cytokine-mediated β-cell apoptosis and is implicated in autoimmune-mediated β-cell destruction, the effect of inhibition of ER stress was studied as a means of cytoprotection in an allogeneic model of islet transplantation [93]. The ER chaperone, glucose-regulated protein 78 (GRP78), is essential for insulin biosynthesis and enhancing ER chaperone capacity was found to improve β-cell function in the presence of prolonged hyperglycemia [94]. Bax inhibitor 1 (BI-1), an evolutionary conserved ER-membrane protein, was found to be a novel modulator of the obesity-associated alteration of the unfolded protein response [95]. Applying cell penetrating peptide technologies, overexpressing the GRP78 and BI-1 rescued high glucose-induced suppression of insulin expression and improved glucose-stimulated insulin secretion with relief of the ER stress. Treatment of GRP78 or BI-1 decreased insulitis and improved glucose metabolism in NOD mice [93]. Both ex vivo and in vivo treatment of GRP78 or BI-1 prolonged islet graft survival after syngeneic islet transplantation, with a higher preservation of the engrafted endocrine tissue and reduced inflammation. Importantly, a longer delay in the allograft rejection after transplantation of GRP78 transduced islets was achieved when mice were also treated in vivo. These may implicate the contribution of ER stress in the allograft rejection process and the permissive role of the β-cell defect may be extended into T1D pathogenesis at large.

Interestingly, the pancreas of patients with T1D are reported to be smaller than those from unrelated controls. Although the pancreas of at-risk individuals are similar in size to that in patients with T1D, the pancreas decreases in size with disease progression [96,97,98]. It is well-known that small β-cell mass might lead to less β-cell functional capacity, which will cause increased pressure on β-cells to cope with adequate glycemic control. In addition to metabolic stress, viral infections or intestinal inflammatory products such as cytokines may leak into the pancreas, which would create a pro-inflammatory environment. This way, β-cells are exposed to viral infection as they express specific receptors and adhesion molecules. Indeed, the presence of a coxsackie virus and adenovirus receptor that is unique to β-cells, found in the insulin-containing granules, might leave β-cells vulnerable to viral infection during insulin secretion, as demonstrated in studies of the association between enteroviral infection by coxsackie virus B4 with islet autoimmunity [99,100]. Although viral infection might be a risk factor in only a small portion of T1D patients, there is a minor but definite supplementary role of viral contribution to the development of T1D and may be extended to cytomegalovirus or even rotavirus [101].

With a similar stand, the loss of coordinated insulin production and secretion from pancreatic β-cells is a key pathological feature of T1D. Increased circulating levels of incompletely processed insulin (proinsulin) are commonly observed in T1D [102]. Intracellular calcium ion (Ca^2+^) stores regulate the biosynthetic burden of insulin production and maturation and the membrane pump sarco/endoplasmic reticulum Ca^2+^ ATPase (SERCA) moves Ca^2+^ through the cells to facilitate the maturation of proinsulin to insulin and insulin secretion. Interestingly, researchers have generated β cell-specific SERCA2 deletion (βS2KO) mice and found age-dependent glucose intolerance and increased serum and pancreatic levels of proinsulin [103]. ER Ca^2+^ levels and glucose-stimulated Ca^2+^ synchronicity were reduced in these βS2KO islets. Reduced SERCA2 expression found in NOD mice dysregulates ER Ca^2+^ release, a common pathway of β-cell dysfunction in T1D. In this animal model, the impaired glucose secretion is accompanied by an increase in proinsulin in the pancreas and in serum [103]. In fact, a SERCA deficiency leads to the reduced production of prohormone convertase, reduced conversion, and an accumulation of proinsulin in the ER–Golgi complexes. Increased circulating proinsulin precedes the first symptoms of T1D by about 12 months [102]. Therefore, impaired proinsulin processing may be a useful biomarker or predictor of T1D. Knocking out SERCA2 reduces ER calcium and accelerates the onset of diabetes in NOD mice [103,104]. These same mice also show progressive mitochondrial dysfunction and weaker and more immunogenic β-cells, leading to accelerated antibody development (or T-cell activation). There may also be mitochondria-mediated immune activation in play. On the contrary, activation of SERCA2 reduces the diabetes development in NOD wild-type and SERCA knockout mice [103]. As Teplizumab has been found to delay the progression of T1D, various ways to leverage β-cell stress to inform T1D biomarker development are to be studied. In the transcriptome analyses, stressed β-cells moving into T1D upregulate all the usual suspect proteins including protein disulphide isomerase A1 (PDIA1) and nucleobindin 2 (NUCB2). PDIA1/prolyl 4-hydroxylase subunit β (P4HB) is needed for efficient proinsulin maturation and β-cell health in diet-induced obesity animal models and PDIA1 expression is increased in human diabetes. Early animal and human data suggest that serum PDIA1 might be a another useful T1D biomarker [104,105].

Taking various ways of β-cell stress contributing to T1D development into account, these might also be explained by genetic risk of T1D, partly determined by certain genetic variants affecting β-cell health, vitality, and self-defense [26,96]. In fact, β-cell mass and function might have been declining for more than 10 years before clinical manifestation of T1D, adding to increasing metabolic stress in β-cells and vulnerability to autoimmune insults [81,106].

## 11. Role of Diet and Microbiota

Dysbiosis of the gastrointestinal tract elicited by changes in intestinal microbiota, especially an increased Bacteroidetes to Firmicutes ratio has been correlated with expression of anti-islet autoantibodies and onset of T1D [107]. Microbiota may shape peripheral immune tolerance, modulating both the migration and differentiation of immune cells to maintain intestinal homeostasis. Furthermore, local inflammation is affected through short-chain fatty acids (SCFAs) generated by gut bacteria from fermentation of non-digestible carbohydrates [108]. SCFAs have a direct effect on T-cell subsets via histone deacetylase inhibition and activation of mTOR and STAT3 signaling, leading to an increased proportion of regulatory T-cells. In addition, SCFAs can also exert anti-inflammatory effect on neutrophils, macrophages, and plasmacytoid dendritic cells via inducing antimicrobial peptides [109]. As animal studies have shown the protective role of these antimicrobial peptides against T1D, SCFAs might be utilized to prevent the cytokine-induced cell death of human islet cells and to improve β-cell function [110]. However, a human clinical trial in patients with longstanding T1D that aimed to restore epithelial integrity by short-term oral butyrate supplementation failed to show any improvement in adaptive or innate immune response [111]. By the way, diet can affect the microbiome and change the risk of T1D. A low-gluten diet can induce favorable changes in the intestinal microbiome of healthy adults, and low maternal gluten intake during pregnancy appears to reduce the development of T1D in the offspring [112,113]. Consumption of various nutrients can shape the structure and activity of the gut microbiota, which in turn produce numerous molecules absorbed and involved in alteration of β-cell function. However, although alterations in gut microbiota are seen in patients with T1D and some data suggest that there is a fecal-microbiome signature associated with at-risk groups; confirmation of these bacterial signatures in different geographical regions controlling for environmental factors is required to determine their clinical significance and use for diagnostic purposes. It remains to be established whether individuals at risk of T1D or patients newly diagnosed would benefit from a low-gluten diet.

## 12. Etiologic Heterogeneity and Future Prospect (Beyond Utilizing Biomarkers)

The incidence of T1D in different parts of the world varies considerably, suggesting the involvement of genetic as well as environmental etiologies. A different genetic susceptibility or different genetic and environmental interactions might be involved in the etiology of T1D and autoimmunity. In addition, there may be atypical forms of diabetic patients in some parts of the world, and they have not yet been well described. Nonetheless, the typical T1D patient in Asian young age-groups is believed to be an autoimmune disease resulting from the destruction of pancreatic β-cells [30,54,59]. Considerable evidence supports an autoimmune etiology for the development of T1D in this group. Around 80% to 90% of newly diagnosed patients with T1D have islet autoantibodies [114], the prevalence of which decreases with the duration of the disease. T1D in this group is also characterized by emerging antibody levels, much higher before overt disease, and/or by the appearance of multiple islet-associated autoantibodies suggesting epitope spreading. In addition, immune responses to islet auto-antigens can be used as a predictor for impending diabetes among patients with susceptible genetic HLA haplotypes or high genetic risk scores for T1D [62,63]. Those who are classified as LADA may also be T1D patients initially presenting with T2D phenotypes. Therefore, we can find candidates for the future development of T1D among autoantibody-positive LADA patients [60]. Notwithstanding, the main current epidemic of diabetes in this pediatric age is a result of surges in T2D incidence. As the general population becomes more obese (and ethnically diverse), reliance on phenotypic characteristics for differentiating between these types of diabetes is becoming increasingly difficult. Nevertheless, the recognition of differences in treatment strategies, associated disorders, family history, and duration of diabetes and cardiovascular outcomes supports the importance of diagnostic efforts to make a distinction between diabetes types at an earlier stage. Initially, to determine the diabetes type, investigation should include ascertaining the presence of autoimmune dysfunction, as well as an assessment of β-cell function (Figure 1). However, it is generally not possible to be certain of diabetes type at the time of diagnosis, and, therefore, initial treatment decisions must be made based on aspects of the presenting physiology, with adjustments in treatment approach made as the individual’s course proceeds and additional information becomes available (Figure 2).

The apparent overlap between T1D and T2D that occurs in obese young patients has resulted in some controversy regarding mixed forms of diabetes, but this does raise unsettling questions about the treatment of T1D in the presence of an insulin-resistant phenotype. There is an etiologic heterogeneity of diabetes even in young ages in Asia. Contrary to Caucasians, the relative proportion of T1aD in young diabetic patients is rather low in Asia. Accepting that the data are very limited, it appears that there is a varying incidence of a T1bD, such as fulminant type 1 diabetes (FT1D) or virus-induced T1D. FT1D exists as a hyper-acute subtype of T1D that affects older children, without causing autoimmunity. They showed a complete loss of β-cell secretory capacity without evidence of recovery, necessitating long-term treatment with insulin (Figure 1). By the way, it also appears that very little is known about the LADA prevalence, especially in Asia. The early diagnosis of LADA or FT1D by appropriate screening methods, such as anti-islet autoantibodies as well as fasting c-peptide concentration in sera of T2D patients, can enhance treatment and potentially lead to methods to slow down the process of destruction against pancreatic β-cells with a possibility to delay or prevent insulin dependency. In addition, practitioners may want to include screening with FPIR in IVGTT and/or its alternatives as well as serum proinsulin concentration (Figure 1) [80,81,82,103]. Distinguishing T1D, T2D, and other forms of diabetes in patients with a younger onset is challenging in Asian populations, because the correct diagnosis is clinically important and has implications for prognosis and management. Although reliable biomarkers that can accurately diagnose and stage T1D across the entire disease spectrum do not yet exist, a diagnostic biomarker, in conjunction with a prognostic biomarker, would allow the identification of high-risk individuals on whom resources should be concentrated. Despite etiological heterogeneity, in the usual clinical setting, however, early diagnosis and classification of patients with diabetes relying on clinical grounds including measuring islet autoantibodies and fasting plasma c-peptide could be a possible viable method to minimize complications [59].

By the way, recent progress in a methodology of diabetes therapy resulting in a near normoglycemic state in many T1D patients irrespective of their subtypes needs to be mentioned. Recently, a method of monitoring glycemia called continuous glucose monitoring system (CGMS) has been utilized in the care of T1D patients. CGMS might be useful in monitoring the glucose profile and detecting hypoglycemic episodes. This also allows for the verification of the meal dose of insulin and a decrease in postprandial hyperglycemia. A standardized approach to CGMS data collection and reporting has been developed and used in clinical trials and these experiences may encourage the use of these metrics in the everyday care of T1D. These will enhance the interpretability of CGMS data, which could provide useful information other than A_1c_ for informing therapeutic and treatment decisions [115]. In addition to CGMS, intermittently viewed CGM devices for at-home patient use with minimally invasive devices have become available [116]. Both CGM and intermittently viewed CGM allow the examination of glucose concentration patterns over time and obviate the need for frequent capillary blood-glucose measurements (Figure 2). When CGM is incorporated into hybrid closed-loop insulin-pump systems that automatically regulate basal infusion rates, but that require manual delivery of meal boluses by trained wearers to cover estimated carbohydrate intakes, substantial improvements in glucose variability and overall glycemic control can be pursued. The substantial advances that have been made in pump and sensor technology and the increase in the number of trials to test their efficacy show that partially or fully automated systems could become a reality. Despite etiological heterogeneity, advances in these methods of diabetes therapy including the detailed monitoring of glycemia in the earlier period of diagnosis may obviate or lessen the possible needs of early diagnosis and classification of young patients with diabetes (Figure 2). Utilizing CGMS in the earlier period of diagnosis could also be a possible viable method to minimize complications.

## Figures and Tables

**Figure 1 ijms-26-03249-f001:**
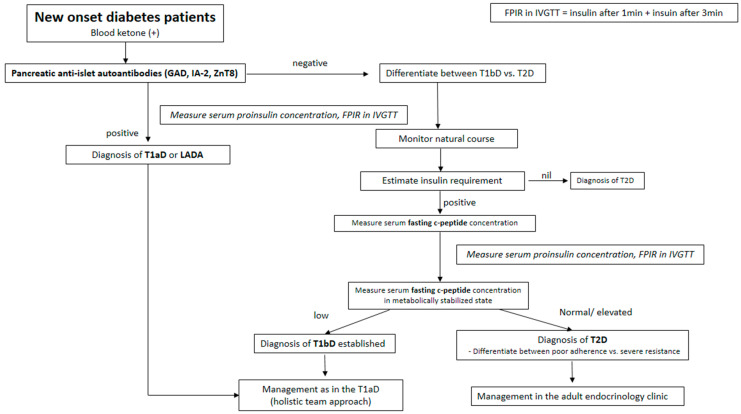
A universal guide to approach to the determination of diabetes type in Asian patients with pediatric age.

**Figure 2 ijms-26-03249-f002:**
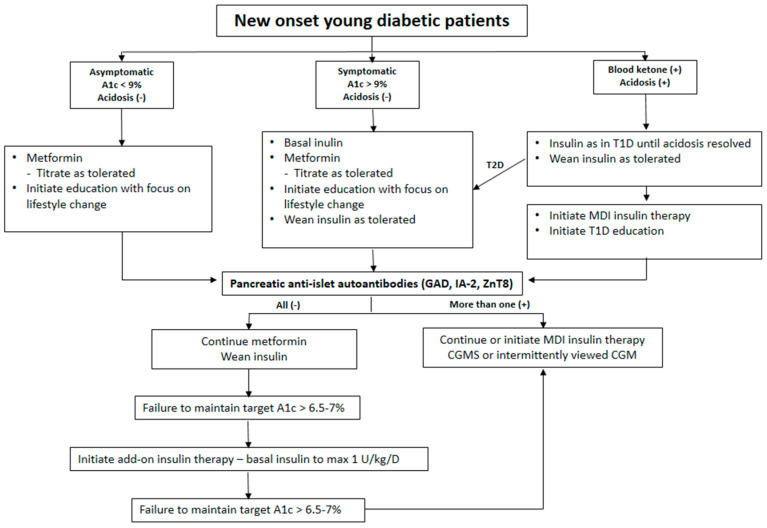
A universal practical guide to an approach to the determination of diabetes type taking natural history into account in young new-onset diabetic patients.

**Table 1 ijms-26-03249-t001:** Etiological Classification of diabetes mellitus (American Diabetes Association).

	Previous Designations	Etiological Distinctions	Clinical Distinctions
**Type 1 DM:** (a) Immune-mediated (b) Idiopathic	(a) Juvenile onset IDDM	Beta-cell destruction (a) Autoimmunity (b) Unknown	Both (a) and (b) result in insulin dependence with loss of β-cells
**Type 2 DM**	NIDDM	Insulin resistance plus relative insulin deficiency	Early in the disease oral hypoglycemic therapy is effective
**Other specific types**	Secondary diabetes	Specific genetic defects MODY* 1, 2, 3, 4, 5 Pancreatic disease Endocrinopathies Chemical-induced Infection-related Immune-mediated forms Genetic syndromes	
**Gestational DM**	Unchanged	Onset during pregnancy	

* MODY: maturity-onset diabetes of the young, DM: diabetes mellitus, IDDM: insulin dependent diabetes mellitus, NIDDM: Non-insulin dependent diabetes mellitus.

**Table 2 ijms-26-03249-t002:** Crucial anti-islet autoantibody assays.

Antigen	Sensitivity (Specificity)	Comment
Insulin *	40–95% (99%)	Inversely age of diabetes onset related
GAD65 *	70% (99%)	Glutamate decarboxylase 65, Predominantly age independent
IA-2 *	60% (99%)	insulinoma antigen 2 (Islet protein tyrosine phosphatase)
Zinc transporter 8 *	60% (99%)	
Chromogranin A		
IGRP		islet-specific glucose 6 phosphatase catalytic subunit-related protein

* During the development of T1D, seroconversion of islet autoantibodies to insulin, GAD65, IA-2 or zinc transporter 8 represents the first notable sign of autoimmunity and their combined presence in serum remains the best predictor for both autoimmunity and T1D.

**Table 3 ijms-26-03249-t003:** High-risk and protective HLA haplotypes found in Asians.

**High Risk**			
DR3	DRB1*0301	DQA1*0501	DQB1*0201
DR4	DRB1*0401	DQA1*0301	DQB1*0302
	DRB1*0402	DQA1*0301	DQB1*0302
	DRB1*0405	DQA1*0301	DQB1*0302
**Moderate Risk**			
DR1	DRB1*01	DQA1*0101	DQB1*0501
DR8	DRB1*0801	DQA1*0401	DQB1*0402
DR9	DRB1*0901	DQA1*0301	DQB1*0303
DR10	DRB1*1001	DQA1*0301	DQB1*0501
**Protective**			
Strong protection			
DR2	DRB1*1501	DQA1*0102	DQB1*0602
DR5	DRB1*1101	DQA1*0501	DQB1*0301
Weak protection			
DR4	DRB1*0401	DQA1*0301	DQB1*0301
DR4	DRB1*0403	DQA1*0301	DQB1*0302
DR7	DRB1*0701	DQA1*0201	DQB1*0201

**Table 4 ijms-26-03249-t004:** The best presumptive non-HLA Genes.

Gene	SNP	Function	Allele	Others (Function/Onset Age/Replication in Asians)
*INS*	rs3842753	Insulin	G	Central immune tolerance, early onset, replication in Asians
*PTPN22*	rs2476601	Protein tyrosine phosphatase	A	Peripheral immune tolerance, not polymorphic in Asians
*ERBB3*	rs2292239	Erb-b2 receptor tyrosine kinase 3	A	Cytokine-induced β cell apoptosis, late onset, replication in Asians
*PTPN2*	rs254215	Protein tyrosine phosphatase, non-receptor type 2	G	Innate immune response, early onset, replication in Asians
*BACH2*	rs72928038	BTB domain and CNC homolog 2, Basic leucine zipper transcription factor 2	A	Immune-mediated β cell apoptosis, late onset, replication in Asians
*UBASH3A*	rs11203203	Ubiquitin-associated and SH3 domain-containing protein A	C	Regulates NF-κB signaling in T cells, early onset, not replicated in Asians
*C1QTNF6*	rs229541	C1q and tumor necrosis factor related protein 6	A	Enterovirus infection, innate immunity, not replicated in Asians
*CLEC1*	rs10492166	C-type lectin domain family 1	A	Dendritic cell activation, Th17 immune response, not replicated in Asians
*IFIH1*	rs2111485	Interferon-induced helicase	A	Enterovirus infection, innate immunity, not replicated in Asians
*CTLA4*	rs3087243	Cytotoxic T lymphocyte-activated 4	A	Regulatory T-cell, early onset, replication in Asians
*CLEC16A*	rs12708716	C-type lectin domain family 16 A	G	Thymic epithelial cell autophagy, mitophagy, early onset, replication in Asians
*TYK2*	rs2304256	TYK2 kinase	T	Vulnerable Beta cells, late onset, not replicated in Asians
*IL2RA*	rs41295121	Interleukin 2 receptor alpha, CD25	T	Regulatory T-cell, early onset, replication in Asians
*GLIS3*	rs7020673	The transcription factor Gli-similar 3	A	β cell development, insulin gene expression regulation, late onset, not replicated in Asians
*IL27*	rs9924471	Interleukin gene	A	Dendritic cell activation, Th17 immune response, not replicated in Asians

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
