# Peer review of "New Perspectives in Studying Type 1 Diabetes Susceptibility Biomarkers"

_ijms, 2025, doi:10.3390/ijms26073249_

Round 1
Reviewer 1 Report
Comments and Suggestions for Authors
This manuscript entitled “New perspectives in studying type 1 diabetes susceptibility biomarkers” by Park Y et al. is an extensive review regarding autoantibodies and genetic susceptibility genes of type 1 diabetes. Reviewer’s comments are as follows:
- Although authors focus on the autoantibodies that are currently employed for routine assays, discussion about other autoantibodies such as chromogranin, IGRP or proinsulin is needed to understand the pathogenesis of type 1 diabetes.
- To further underscore the clinical utility of autoantigens, vaccination based on the autoantigens or immunotherapy of type 1 diabetes needs to be discussed.
- It is recommended to discuss about immunotherapy of type 1 diabetes using anti-CD3 antibody, anti-TNFa antibody or JAK1 inhibitors, which will be helpful to follow the current status of immunotherapy against human type 1 diabetes.
- Clec16A is known to affect mitophagy. This point needs to be discussed.
Author Response
Thank you very much for taking the time to review this manuscript. Please find the detailed responses below and the corresponding revisions/corrections highlighted/in track changes in the re-submitted files.
- Although authors focus on the autoantibodies that are currently employed for routine assays, discussion about other autoantibodies such as chromogranin A, IGRP or proinsulin is needed to understand the pathogenesis of type 1 diabetes.
Thank you for your suggestion. The authors agreed to follow your suggestion and added a brief discussion on these as follows.
The β-cell molecules, chromogranin A, islet-specific glucose 6 phosphatase catalytic subunit-related protein (IGRP) and proinsulin are autoantigens in the non-obese diabetic (NOD) mouse and help us to understand the pathogenesis of T1D. But robust assays for autoantibodies are not available.
T cells also target chromogranin A and IGRP, which extend the list of T cell antigens in T1D. T1D is not at all or rarely developed in chromogranin A gene knock out NOD mouse.13 Antigen-specific immunotherapy targeting IGRP could induce antigen-specific CD8+ T-cell-targeted immune regulation and delay development of T1D.14
- To further underscore the clinical utility of autoantigens, vaccination based on the autoantigens or immunotherapy of type 1 diabetes needs to be discussed.
Thank you for your suggestion. We agreed to follow your suggestion and added a brief discussion on vaccination based on autoantigens (antigen-based immunotherapy).
T cells also target chromogranin A and IGRP, which extend the list of T cell antigens in T1D. T1D is not at all or rarely developed in chromogranin A gene knock out NOD mouse.13 Antigen-specific immunotherapy targeting IGRP could induce antigen-specific CD8+ T-cell-targeted immune regulation and delay development of T1D.14
In contrast, proinsulin expression from gestation until weaning was sufficient to completely protect NOD mice from diabetes, insulitis, and development of insulin autoantibodies. Inducing tolerance to islet antigens in the perinatal period is sufficient to impart lasting protection from diabetes.76 Autoantigen treatment to prevent T1D in recent onset T1D patients was insufficient in human subjects, but preserved residual b-cell function.75 Besides the possible minor efficacy of peroral insulin in high-risk individuals, autoantigen prevention trials have failed. Adequate autoantigen selection, doses and route would be important.74,77 The overall aim of b-cell antigen vaccination is to induce tolerance by inducing autoantigen-specific T regulatory cells, which also downregulates the activity of proinflammatory antigen-presenting cells.73
- It is recommended to discuss about immunotherapy of type 1 diabetes using anti-CD3 antibody, anti-TNFa antibody or JAK1 inhibitors, which will be helpful to follow the current status of immunotherapy against human type 1 diabetes.
Thank you for your suggestion. We agreed to follow your suggestion and added a brief discussion on immunotherapy.
To regulate the autoimmune response, researchers tried to delete lymphocyte subsets and/or re-establish immune tolerance via activation of regulatory T cells. The use of broad immunosuppressive drugs was the first approach. Subsequently, antibody-based immunotherapies failed to discriminate between autoreactive versus non-autoimmune effectors. Antigen-based immunotherapy is a third approach developed to manipulate β cell autoimmunity, which allows the selective targeting of disease-relevant T cells, while leaving the remainder of the immune system intact.
The anti-CD3 monoclonal antibody teplizumab, which was approved by the US Food and Drug Administration in 2022 to delay the progression of T1D in patients at high risk of developing T1D. Several mechanisms of targeting CD3 with anti-CD3 monoclonal antibodies have been postulated including direct inactivation of T cells, inhibition of apoptosis induction and induction of apoptosis or anergy of activated T cells. While these new therapies show promise, achieving long-lasting control of blood sugar levels and the possibility of stopping insulin use remains difficult.77 Furthermore, immunotherapies targeting cytokines in T1D have not yet proven a considerable effect on disease protection.78,79
- Clec16A is known to affect mitophagy. This point needs to be discussed.
Thank you for your suggestion. We agreed to follow your suggestion and added a brief comment of Clec16A to affect mitophagy in Table 4.
Reviewer 2 Report
Comments and Suggestions for Authors
I have read with great interest the intriguing paper entitled: "New Perspectives in Studying Type 1 Diabetes Susceptibility Biomarkers" which examines the pathogenesis of diabetes and the role of many factors, especially genetic but also acquired.
In my opinion, the paper could be further ameliorated and it is possible to enhance the quality and the impact of the manuscript.
Selection Bias: The paper focuses predominantly on specific genetic markers, such as HLA haplotypes, which are well-studied but may not represent the full spectrum of genetic factors influencing T1D. This could lead to an underrepresentation of other important genetic or environmental factors that contribute to the disease.
Confirmation Bias: During the paper, there is a tendency to emphasize findings that support existing theories about the role of HLA in T1D susceptibility while potentially downplaying or overlooking contradictory evidence. For instance, while the paper discusses the strong associations between certain HLA alleles and T1D, it may not adequately address studies that show weaker or no associations. Please, modify this approach.
Geographic Bias: The paper mentions genetic results from specific populations, such as Korean individuals, which may not be generalizable to other ethnic groups. This geographic bias could limit the applicability of the findings to a broader population, as genetic susceptibility to T1D can vary significantly across different ethnicities. It is important that Authors tell this bias in the discussion.
Focus on Autoimmunity: While the paper acknowledges the role of β-cell dysfunction and apoptosis, it may place too much emphasis on autoimmune aspects of T1D. This could overshadow other critical factors, such as metabolic stress or environmental influences, that also play a significant role in the disease's pathogenesis.
Clarify the Role of Diet and Microbiota: The paper discusses also the possible influence of diet and gut microbiota on T1D. It would be beneficial to provide a clearer explanation of how specific dietary components and microbiota alterations can affect immune responses to β-cell function. This could include examples of beneficial diets or specific microbiota profiles that have been associated with T1D prevention or management.
Author Response
Thank you very much for taking the time to review this manuscript. Please find the detailed responses below and the corresponding revisions/corrections highlighted/in track changes in the re-submitted files.
- Selection Bias: The paper focuses predominantly on specific genetic markers, such as HLA haplotypes, which are well-studied but may not represent the full spectrum of genetic factors influencing T1D. This could lead to an underrepresentation of other important genetic or environmental factors that contribute to the disease.
Thank you for your suggestion. The authors agreed to follow your suggestion and added some discussions on this aspect emphasizing the contribution of non-HLA genes as well as environmental factors.
Actually, although authors elaborated more on HLA haplotypes in explaining genetic factors influencing T1D since HLA is the major gene, we already tried to give a detailed thorough explanation of non-HLA genes in the original version of the manuscript. The importance of non-HLA genes was also shown as we introduced the ‘genetic risk score’, which is a combination of contribution from both HLA & non-HLA genes. We also wanted to put an extra emphasis on explaining environmental factors which are contributed to the disease susceptibility. However, we have the data suggesting that variability of T1D incidence worldwide correlated with prevalence of HLA susceptible haplotypes, though (Park Y. Why is type 1 diabetes uncommon in Asia? Ann NY Acad Sci 2006; 1079; 31-40). The frequencies of T1D‐associated HLA antigens correlate well with T1D incidence rates worldwide. Generally, the DR3 and DR4 haplotypes are common in Caucasians, which means that the prevalence of T1D is likely to be higher due to easy formation of the DQ 0201/0302 dimer. Conversely, the frequency of the DRB1 genotypes among Asian T1D patients points to a much lower contribution of DR3 and DR4, especially the DR3/4 genotype. In East Asians, it appears, moderate‐risk haplotypes are more frequent whereas susceptible haplotypes are rare. As the contribution of HLA DR3/4 is relatively minor, it may be just the tip of an iceberg, with many more minor genes (including non‐HLA) and less frequent genotypes at play. Moreover, the low‐incidence rate in the East Asian population may be explained by the counterbalancing effect of the DRB1 and DQB1 alleles in the control population.
Following sentence was added to avoid underrepresentation of environmental factors that contributed to T1D.
By the way, diet can affect the microbiome and change the risk of T1D. A low gluten diet can induce favorable changes in the intestinal microbiome of healthy adults, and low maternal gluten intake during pregnancy appears to reduce development of T1D in the offspring. 114,115 Consumption of various nutrients can shape the structure and activity of the gut microbiota, which in turn produce numerous molecules absorbed and involved in alteration of β-cell function.
- 2. Confirmation Bias: During the paper, there is a tendency to emphasize findings that support existing theories about the role of HLA in T1D susceptibility while potentially downplaying or overlooking contradictory evidence. For instance, while the paper discusses the strong associations between certain HLA alleles and T1D, it may not adequately address studies that show weaker or no associations. Please, modify this approach.
Thank you for your suggestion. We agreed to follow your suggestion and added a brief discussion on contradictory evidence.
Although there have been some studies that show weaker or no association, several studies have demonstrated that both DQ and DR influence T1D susceptibility.29,30
Although the effects of these alleles are the same across different countries, ethnic differences in the prevalence of specific alleles in patients varies with population allele frequencies, which makes different level of association.
However, genetic susceptibility to T1D can vary significantly across different ethnicities and importance and strength of individual non-HLA genes may differ.
The incidence of T1D in different part of the world varies much, suggesting the involvement of genetic as well as environment etiologies. In addition, there may be atypical forms of diabetic patients in some part of world and they have not yet been well described. As we described a universal guideline to approach to the determination of diabetes type in Asian patients, we tried to abide by this rule and make a general guideline taking heterogeneity of T1D into account.
- Geographic Bias: The paper mentions genetic results from specific populations, such as Korean individuals, which may not be generalizable to other ethnic groups. This geographic bias could limit the applicability of the findings to a broader population, as genetic susceptibility to T1D can vary significantly across different ethnicities. It is important that Authors tell this bias in the discussion.
Thank you for your suggestion. We agreed to follow your suggestion and if the data are from Korean or Asian, show them as such. We also added a brief discussion on this matter and following sentences were added.
However, susceptibility to T1D can vary significantly across different ethnicities and importance and strength of individual non-HLA genes may differ.
Although we have a different opinion about the differential genetic susceptibility across different ethnicities, when we used Korean or Asian data for discussion, we narrowed down its meaning and showed them as ‘Korean’ or ‘Asian’. In fact, all the genetic results we mentioned are originated from Caucasian data at first and then moved to the Korean data.
Although the risk associated with individual HLA DR or DQ alleles appears to vary between populations, when comparing identical DRB1‐DQB1 haplotypes, the association and transmission to diabetic offspring is similar for Koreans and Caucasians, suggesting that these haplotypes transcend ethnic boundaries (Park Y, She JX, Wang CY, et al. Common susceptibility and transmission pattern of HLA DRB1‐DQB1 haplotypes to Korean and Caucasian patients with type 1 diabetes. J Clin Endocrinol Metab. 2000;85:4538‐ 4542).
However, a different genetic susceptibility or different genetic and environmental interactions might be involved in the etiology of T1D and autoimmunity.
- Focus on Autoimmunity: While the paper acknowledges the role of β-cell dysfunction and apoptosis, it may place too much emphasis on autoimmune aspects of T1D. This could overshadow other critical factors, such as metabolic stress or environmental influences, that also play a significant role in the disease's pathogenesis.
Thank you for your suggestion. We agreed to follow your suggestion and added a brief description of ER stress in the beginning of the text of the subtitle, ‘T1D as a β-cell disease’.
Normal β cell physiology includes a high demand for insulin production and secretion in response to dynamic glucose sensing. This secretory function predisposes β cells to significantly higher levels of endoplasmic reticulum (ER) stress compared to nonsecretory cells. In addition, many environmental triggers associated with T1D onset further augment this inherent ER stress in β cells.
However, most part of the abstract were already covering β-cell aspects. We also elaborated on metabolic stress especially ER stress quite in detail in the main text, especially in the text of the subtitle, ‘T1D as a β-cell disease’. We also described environmental influences including the data of virus much, though. However, the classic view of T1D pathogenesis are to be introduced in the beginning.
- Clarify the Role of Diet and Microbiota: The paper discusses also the possible influence of diet and gut microbiota on T1D. It would be beneficial to provide a clearer explanation of how specific dietary components and microbiota alterations can affect immune responses to β-cell function. This could include examples of beneficial diets or specific microbiota profiles that have been associated with T1D prevention or management.
Thank you for your suggestion. We agreed to follow your suggestion and added a brief discussion on the role of diet and microbiota.
By the way, diet can affect the microbiome and change the risk of T1D. A low gluten diet can induce favorable changes in the intestinal microbiome of healthy adults, and low maternal gluten intake during pregnancy appears to reduce development of T1D in the offspring. 112,113 Consumption of various nutrients can shape the structure and activity of the gut microbiota, which in turn produce numerous molecules absorbed and involved in alteration of β-cell function. It remains to be established whether individuals at risk of T1D or patients newly diagnosed would benefit from a low gluten diet.
- Hansen LBS, et al. A low-gluten diet induces changes in the intestinal microbiome of healthy Danish adults. Nat Commun 2018; 9: 4630.
- Antvorskov JC, et al. Association between maternal gluten intake and type 1 diabetes in offspring: national prospective cohort study in Denmark. BMJ 2018; 362: k3547.